# E-Commerce in Agri-Food Sector: A Systematic Literature Review Based on Service-Dominant Logic

**Mengzhen Zhang \* and Sami Berghäll**

Department of Forest Sciences, University of Helsinki, 00014 Helsinki, Finland; sami.berghall@helsinki.fi
\* Correspondence: mengzhen.zhang@helsinki.fi

**Abstract:** Agricultural e-commerce (AE) has attracted substantial attention within various research disciplines for several years. In this paper, we present a literature review of the recent state of AE research published from 2000 through to 2021 in 83 journals. Based on Service-Dominant Logic (S-D logic) and Qualitative Comparative Analysis (QCA), we identify six research themes, and a theoretical continuum is applied to reveal how research themes and scholarly approaches fit into the S-D logic framework. A general increasing trend in the number of articles confirms the escalating interest in AE research; however, different themes perform unevenly with S-D logic. Even though research themes such as Consumer Willingness are getting closer to S-D logic premises, and ideologies that are increasingly approaching S-D logic have been applied to analyzing AE topics, unfortunately, there remains a paucity of papers that wield S-D logic in the AE field. Our research focuses on an innovative emerging AE field and, simultaneously, provides an approach of integrating S-D logic into analyzing academic papers in the AE domain. This research may shed some light on future possibilities that S-D logic could support the co-creation of value between consumers and agribusiness managers, and other broader disciplines such as management and marketing.

**Keywords:** agricultural e-commerce (AE); customer relationship management; value co-creation; service-dominant logic (S-D logic); literature review

## 1. Introduction

E-commerce is developing rapidly and has penetrated almost all sectors. Agriculture is identified as being promising due to its high level of fragmentation [1]. The benefits of E-commerce, inter alia, are that it boosts the circulation of agricultural products and development [2], enables smoother communication and better experiences [3], and promotes market transparency and price discovery [4]. To sum up, e-commerce brings various benefits to the agricultural sector, which have been promisingly predicted for potential success in AE field.

Customer relationship management is a critical factor in e-commerce [5]. Customer relationship management provides an opportunity to create loyal e-commerce consumers who make repeat purchases [6], as increasing purchase intention is an essential goal when it comes to e-commerce success [7]. Researchers have sought to understand the connections between consumers and e-commerce, and the main findings suggest there is an interplay effect. On the one hand, e-commerce has enhanced the efficiency of interaction with consumers [8]. Furthermore, it provides firms with the ability to reach new customers, as well as old customers, in new ways [9]. On the other hand, e-commerce is a convenient way of obtaining products and services [10], also leads to greater customer satisfaction [11], and is expected to attract more consumers and increase demand [12] of a company's products.

In connecting e-commerce with agriculture, Zeng et al., (2017) [13] are one of the first in building a systematic literature review on agricultural e-commerce (AE). However, their review is focused on factors affecting firm-level adoption and regional development of AE, leaving customer-related themes in the sidelines. Other authors also conduct research

about potential reasons for consumer reservations to buy agricultural products online. The reasons found are quality and service concerns [14]. The nature (fragility, perishability, etc.) of agri-food products is also seen as one of the main reasons slowing down e-commerce adoption in agriculture (G. Baourakis et al., 2002) [15]. However, due to a relatively new state of e-commerce in agriculture, its impact has not been widely measured and documented [16], and thus, there remains a need to proceed with comprehensive research on consumer behaviors of e-commerce in the agricultural environment [17]. Therefore, we argue that there is a gap in the disciplinary research of customer–company interaction related to agricultural e-commerce. This gap exists especially in modern conceptualization of value co-creation and the core philosophical nature of the phenomenon itself. We use the Service-Dominant Logic (S-D logic) developed by Vargo and Lusch (e.g., 2004, 2016, 2018) to evaluate the agricultural e-commerce literature and its approach to company–customer interaction.

Replacing what Vargo and Lusch (2004) label goods dominant logic with S-D logic, we obtain a new perspective understanding the set of processes through which value is created and delivered to the consumer. The key is the notion that the customer should be considered as a major element in the value co-creation process. While some researchers have applied S-D logic to understand value creation in AE field—for instance, Xiaoping et al., (2009) [18] and Carpio et al., (2013) [19]—there is still a lack of comprehensive discussion of how S-D logic has been applied to the AE domain and what are possible new angles that S-D logic could provide us with. Thus, this paper makes an effort to bridge some of the above-mentioned gaps in the scholarly literature. We do this via a literature review where the scholarly papers are classified into classes based on key S-D logic dimensions. As data, we use the published AE journal articles existing in the Scopus and Web of Science databases. In line with Webster and Watson (2002) [20], we choose literature review as an approach as it creates a foundation for advancing knowledge, facilitates theory development, and uncovers areas where research is needed. A literature review also allows researchers to build a basic understanding of where the scholarly discussion is going (ibid.). Therefore, this paper tries to answer the following questions:

(RQ1) What are the major research themes and current focuses in the already published papers within the domain of AE?

(RQ2) How do the research themes and scholarly approaches fit into the S-D logic framework?

(RQ3) What possible future directions need to be drawn from the previous two?

The paper proceeds by introducing a discussion of S-D logic, followed by analyzing the research themes, keywords and theories prevailing in the existing literature. We then design a classificatory tool to answer the above questions. In the end, we discuss the conclusions that can be drawn from the analysis.

## 2. Service-Dominant Logic Explained

Service-Dominant Logic (S-D logic) was first put forward by Vargo and Lusch in a seminal paper in 2004 and, since then, it has been the most cited article in the *Journal of Marketing* in the last decade [21]. From 2004 to 2016, Vargo and Lusch (along with other researchers) jointly published more than 200 articles on S-D logic and its related foundational promises. In this service-dominant view of exchange, tangible goods should be seen merely as vehicles of service provision [22]. Service, or the actual value received from the service, is the basis for market competition, and the customer is seen as an *operant resource* (ibid.). Customer experiences [23] are important and value is phenomenologically born out of exchange settings, but is unique to those involved in the exchange. Thus, the customer should always be regarded as a co-creator of value [24]. In addition, S-D logic treats all customers, employees, and organizations as operant resources, thus seeing them as endogenous to both the exchanges and the value-creation processes [25]. The relationships are not company–customer or producer–user relationships, but Actor-to-Actor (A-to-A) relationships. In conclusion, the core of the theoretical work in the S-D logical sphere is to

build an axiomatic basis for economic exchange in a way that would include how value, social systems, capabilities, and resources explain the birth of prosperity out of human economic relationships [26].

Service-dominant logic opens an insightful view to reconsider the value that the exchange setting brings. S-D logic sets out several foundational premises that comprehensively challenge traditional marketing assumptions [27]. Therefore, while it is seen as good commercial sense to improve customer retention rates [28], S-D logic brings in the customer as a value co-creator in the equation (e.g., Vargo and Lusch, 2004, 2008, 2016 [22,24]).

According to Vargo and Lusch (2004, 2008), a goods-dominant logic (G-D logic) is the prevailing, traditional idea of marketing exchange. G-D logic probes into the process of value creation, and assumes value is created by the firm and distributed through the exchange of goods and money, while the role of consumer is to "use up" or "destroy" the value created by firms [29]. Once the value creation process is pre-defined as an input by the manufacturer, the consumer is regarded as a "receiver" because their role is limited to only receiving the outcomes of the exchange. By contrast, in S-D logic, "producer" no longer has a dominant leading role in value creation. The customer (or beneficiary) is now considered as a major part in the value co-creation process of the exchange setting (Vargo and Lusch 2004, 2008). Secondly, the distinction between "receiver" and "producer" disappears, because value is co-created through the combined efforts of firms, customers and employees, and other entities related to any given exchange setting (ibid.). In other words, market relationships are not built on separate units interacting with each other sporadically but, instead, intertwined tightly when value creation is seen from the S-D logic perspective. Service-dominant logic empowers consumers and highlights the fundamental role consumers play in the market.

Based on the previous discussion, the crucial reason why this paper adopts S-D logic is to understand the core phenomena of the agricultural e-commerce exchange setting. Thus, as e-commerce enables new forms of firm–consumer interaction described by more intense consumer participation [30], S-D logic provides a precise mindset to describe and explain how firms and consumers integrate their resources to co-create value in this relatively new setting. Applying S-D logic in AE context helps us:

(1) To interweave S-D logic into the existing academic literature;
(2) To draw conclusions from the emerging ideas;
(3) To propose possible research trends;
(4) To shift our research focus from G-D logic to A-to-A relationships;
(5) To pay special attention to consumer value creation.

## 3. Material and Research Method

In this section, we will discuss how we conduct our literature review in each step. We used Snyder's (2019) [31] guidelines for conducting a systematic literature review. Based on her study, the first step should be designing the review. The process here is "*select search terms and appropriate databases and decide on inclusion and exclusion criteria*". Therefore, in line with our research topic, three search strings related to e-commerce, agriculture and the consumer are first identified. Keywords related to e-commerce include "electronic commerce" or "electronic business" (Amit and Zott, 2001; Zhu, 2004; Nanehkaran, 2013; Bodini and Zanoli, 2011) [32–35], or "e-commerce" or "e-business" in short. The keywords for agriculture are "agriculture" and "agricultural" (Montealegre et al., 2007 [12]; Ma et al., 2018a [36]), and also "agri-food" and "agribusiness". Considering that FAO (2017) [37] points out that agri-food systems are undergoing a rapid transformation, the term "agribusiness" should thus be highlighted. Other researchers also focus on these two terms, for example, agri-food (Sturiale and Scuderi, 2017a; Fritz et al., 2004) [38,39] and agribusiness (Montealegre et al., 2007 [12]; Becvarova, 2005 [40]).

Keywords "consumer" and "customer" are used interchangeably; see Jansen et al., (2009) [41] and Meuter et al., (2005) [42].

Scopus and Web of Science are examples of numerous databases that are available for researchers to review the main literature in one domain [43]. Web of Science is the major research database for citation tracking, while Scopus (www.scopus.com, accessed on 13 September 2020) has greater coverage of open access and international journals [44]. Scopus does a creditable job in the social sciences but does not "reach back" as far as Web of Science [45]; therefore, this paper incorporates these two databases to avoid missing possible research targets. With these features in mind, the mentioned two are selected as the databases used for this paper. However, because Scopus and Web of Science require different coding formats, final search strings have slight differences. For Web of science, search strings are: ("e-commerce$" OR "electronic commerce *" OR "e-business" OR "electronic business") AND (agriculture OR agricultural OR agrifood OR agribusiness) AND (consumer OR customer).

In the Scopus database, we used two patterns of search strings, which provides a broader scope of search. The searches are named as Scopus-search Terms One and Two. The search strategy is to include as many articles as possible in the beginning, and then exclude some of them by narrowing the scope to the task at hand. Scopus-search Term One is designed as: (TITLE-ABS-KEY (e-commerce OR electronic commerce OR e-business OR electronic business)) AND ((agriculture OR agricultural OR agrifood OR agribusiness)) AND (consumer OR customer) AND (LIMIT-TO (LANGUAGE, "English")) AND (LIMIT-TO (DOCTYPE, "ar") OR LIMIT-TO (DOCTYPE, "cp")). Scopus-search Term Two is (electronic AND commerce) OR (e-commerce) OR (e-business) AND ((agriculture)) OR (agricultural) OR (agrifood) OR (agribusiness) AND ((consumer)) OR (customer).

Scopus-search Term One produces 535 hits initially (accessed on 13 September 2020); there are 154 hits obtained by using Scopus-search Term Two, and 87 hits in Web of Science. The actual analysis is carried out using Mendeley's reference management software. Full-text copies of the articles are obtained and imported into Mendeley.

In the first round, we exclude conference papers considering their inconsistent quality, thus limiting the inquiry into journal articles only. The second-round selection is made after reading each article's abstract, or the full article when necessary. Articles not written in English are excluded. Articles which cannot be downloaded or missing full text are also excluded from the study. A third-round selection is made based on the research topics to see if they are highly related to the AE context. Articles not related to AE at all, and articles only mentioning AE briefly as a research background, with focus more on algorithm or other technical practices, are excluded from the review. For example, Fecke et al., (2018) [46] only use AE as an insight to understand German farmers' intention to use the Internet and, in a similar fashion, so does Warren (2004) [47], who tries to investigate UK farmers' adoption of the Internet in agriculture but who has scarcely any argument from the customer's perspective, etc. These papers present a new perspective by introducing AE into their research; however, their focuses are not AE itself, thus papers of this kind have been deducted.

Finally, there are 106 papers remaining, including 92 articles in Scopus and 14 articles in Web of Science. The reasons for such a loss are that a large proportion of the papers are conference papers with relatively unstable quality, and many of the articles do not focus on agricultural context, etc.

## 4. Description of the Data

The 106 articles spanned from 2000 to 2021. Figure 1 displays the distribution of the research articles across the study period.

A general increasing trend in the number of articles confirms the escalating interest in AE research, with a sharp acceleration appearing after 2013, which suggests that this area is relatively new, but has gained greater attention from academia in recent years.

Based on the number of times the paper has been cited, 5 papers have been cited more than 40 times, 5 papers have been cited between 30 and 40 times, 11 papers have been cited ranging from 20 to 30 times, and the rest of the 85 papers have been cited less than 20 times.

This result indicates that a large proportion of AE papers have not been spread widely and this domain is has relatively limited influence among other research topics, when comparing to the representative S-D logic paper [24], which has been cited 3437 times.

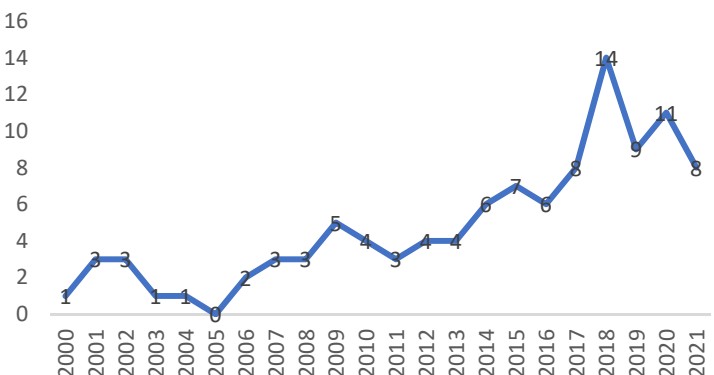

**Figure 1.** Distribution of research articles.

Our analysis shows that these 106 research articles come from 83 journals. The largest sources are *British Food Journal* (5 articles) and *Sustainability* (5 articles). This paper takes the SCImago Journal Rank (SJR), in the year of 2019, as an indicator to evaluate journals' impact. SJR is both wide and dynamic enough to measure the evolution of scientific journals [48], higher SJR values indicating greater journal impact ("SCImago Journal Rank-Wikipedia", n.d.). We find 14 journals (15.7% of the 83 journals) with a SJR value above 1 and there are 16 papers published in these 14 journals, which may indicate that papers published in the AE domain selected journals with high impact, but their quantity is still limited.

Figure 2 displays the keywords that appeared more than 10 times and their percentage found in our research papers. There are five papers lacking keywords; we therefore collected 101 sets of keywords. As can be seen in Figure 2, e-commerce/electronic commerce is the most frequently used keyword, which has been used for 59 times. "food" follows with 22 instances; "Internet" and "model" follow with 17 instances; and "online" appears 15 times. "Business", "marketing", "supply chain" appear 14 times each, indicating that our research is intertwined with the marketing domain and that our research is multi-disciplinary in nature.

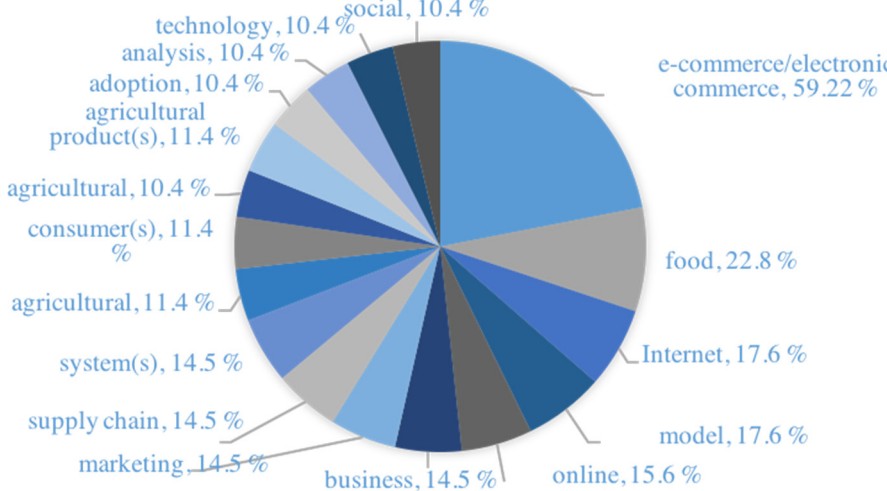

**Figure 2.** Overview of the most frequently used keywords.

**5. Major Themes in EM Research**

*5.1. Six Themes Identified*

To build a distinct understanding for the analysis of the AE literature, the articles are first grouped into different themes based on the abstracts, keywords, as well as introduction when necessary. Our classification uses the system developed by Wang et al., (2008) [48]. In their paper, they code electronic marketplace (EM) research into various themes, EM success and EM impact being two of them. Further, in the theme of EM success, sub-categories are: success factors, reasons of failure, successful strategies of EMs, and EM performance and its antecedents. The theme-coding practice helps them to understand what the topics that have gained researchers' interest are, and what could be the possible trends that demand attention from future research. Holding a similar goal and in line with their research output, we code 106 papers to six research themes, which are quality of AE performance, the historical development of AE (history and influence), opportunities and barriers that e-commerce brings to agriculture, factors influencing AE, possible solutions to solve the problems when encountering barriers, and the terminal goal is how AE affects users' willingness to use AE services (consumer willingness). The following sections describe the basis of this classification.

5.1.1. History and Influence

This section summarizes articles overviewing the history and development of agricultural e-commerce, and also how e-commerce influences agricultural business. For example, Baourakis et al., (2002) [15] introduce the impact e-commerce has on agri-food marketing in Cretan agricultural cooperatives. Canavari et al., (2010) [49] explain technology's role and especially e-commerce's role in the agri-food sector, with special attention paid to trust in B2B relationships in an electronic context. Under this theme, we have an overview of how e-commerce is emerging and penetrating the agri-food market in various countries and understand its effects on the agri-food sector.

5.1.2. AE Performance

Articles evaluating agricultural online business performance are coded in this category. For example, Dal Vecchio et al., (2018) [50] conduct an analysis on the market performance of Italian wines in the Chinese e-commerce market; four Chinese e-commerce platforms, including Taobao, are investigated. Borsellino et al., (2018) [51] evaluate the e-service quality of websites in the extra virgin olive oil sector, and formulate marketing strategies to improve online business performance. Similarly, Galati et al., (2016) [52] assess the websites' quality of Italian wineries while also verifying the existence of a relationship between website quality and business revenue. Sun et al., (2019) [53] take China as an example and evaluate electronic service quality of agricultural business websites from consumers' perspectives. This theme gives readers a brief understanding of how smooth e-commerce operates in agricultural industry and how much AE fits in customers' expectations.

5.1.3. Factors Influencing AE Adoption

Articles listed in this category focus on the factors affecting the development of e-commerce in the agricultural sector. The most frequently used terms in this category are attributes, factors, parameters, elements, variables, criteria, etc. There are 26 articles found in this category. We also count which countries researchers are paying attention to. We find that China has gained the largest amount of attention, as six articles discuss e-commerce-related topics, such as factors influencing the agricultural product marketing and logistics [2], product similarity and recommender systems influencing online shopping outcomes [54], etc. Other countries that are mentioned under this category are: Italy (three articles), Greece (three articles), Spain (two papers), and so on. This finding suggests that AE research might have a distinct area concentration, while China, alongside Italy and Greece, are leading the current research trend in the AE domain.

### 5.1.4. Potential Opportunities and Barriers

E-commerce brings new possibilities and opportunities to traditional industry. Many researchers, therefore, analyze the pros and cons existing in the agricultural e-commerce area. For example, challenges and opportunities that the digital economy presents for agribusiness in India [55]; e-commerce opportunities in the agricultural sector in developing countries such as Tanzania [56]; possible barriers of e-commerce adoption faced by SMEs in Australia and Indonesia concerning the immaturity of e-commerce in agriculture [57]; enablers and barriers to e-commerce in Tanzanian small and medium enterprises (SMEs) and challenges faced by SMEs [58].

### 5.1.5. Possible Solutions

Researchers have conducted a large amount of work in an attempt to suggest feasible solutions to improve the efficiency of agricultural e-commerce. They suggest models, frameworks, or try to build platforms or new systems to boost the development of AE. We found 37 interdisciplinary articles in this category. For instance, a blend of top-down and bottom-up approaches to encourage the diffusion of online marketing activities [59]; a model to analyze the relationships between three aspects of technical electronic commerce (EC)-based information system (IS) resources [60]; a conceptual framework to test SMEs' e-commerce adoption levels in Jordan [61]; policy suggestions on the feasibility of building a mobile e-commerce platform for fresh agricultural products [62], etc. Papers listed in this category are comparatively diverse as researchers are from various specializations, focusing on practical algorithms, policy suggestions, marketing strategies, and other interdisciplinary subjects.

### 5.1.6. Consumer Willingness/Acceptance to Accept E-Commerce

Consumers are important participants and main users in the e-commerce section; therefore, factors influencing consumers' acceptance and willingness to use e-commerce services are another emerging topic. Terms or contexts, for instance, consumer shopping behavior (Wang et al., 2019) [63], customer relationship management [64], or customer satisfaction [65], etc., are directed into this category. The main research findings are that trust and web services [66], product, service quality and price [67], perceived ease of use and usefulness of the website [68] and other factors could all influence customer shopping behavior and satisfaction. Table 1 shows all papers coded under six themes.

### 5.2. Literature-Driven Classification of S-D Logic in AE Research

Similar to Wang (2008), the core logic of identifying themes is to scan existing papers in the EM research to reveal topics already exploited and issues still not fully explored. Aside from these aims, this identification step also helps the authors to further produce a comprehensive explanation interweaved with new perspectives. Our aim in the following section tackles how the papers collected before address the new view on exchange as argued by Vargo and Lusch (2004, 2008, 2016, etc.). To achieve this goal, the key is to build a comprehensive tool to classify the disciplinary papers and see if the articles fit the S-D logic criteria.

**Table 1.** Articles classified into the themes mentioned.

| Themes | History and Influence | Performance | Factors | Opportunities and Barriers | Possible Solutions | Consumer Willingness | Themes |
|---|---|---|---|---|---|---|---|
| Number of papers | 7 | 4 | 26 | 11 | 37 | 22 | Number of papers |
| Brief introduction | History and development of agricultural e-commerce, how e-commerce influence agricultural business | Evaluate agricultural online business performance | Dominant factors impacting the development of e-commerce in agriculture | Potential and opportunities; agricultural e-commerce also has barriers and obstacles | Suggest or propose models or other solutions to improve efficiency or development of agricultural e-commerce | Factors affecting consumers' shopping willingness or satisfaction | Brief introduction |
| Most used terms | Overview, developments, impact, effect | Website quality, evaluation, performance | Attributes, factors, parameters, elements, variables, criteria | Opportunities, barriers, problems | Model, framework, platform, suggestion and improvements | Consumer shopping behavior, customer relationship management, customer satisfaction | Most used terms |
| Descriptive S-D logic concepts | Information flow, co-operatives | Cultural background and customs, consumer preference, supply chain | Attributes, customer satisfaction, innovation, certification, attitudes, awareness, decision making, network, consumer assessment, information exchange, intermediate function, interactivity, information flow | Information exchange, social networks, innovation, participation, perception, customer intention, web ecosystem, user-created value | Objective, business values, collaborative networks, attributes, innovation, branding, sustainable development and sustainable change, effectiveness, cultural values, decision making, relationship quality, interactivity, marketing communication | Consumer relationship management-related topics, including: satisfaction and value, shopping intention, attitudes, demand, loyalty, trust, shopping behavior, service, concerns, security, perceived risk, decision making, attributes, perceptions, social networks, branding, effectiveness, cultural values | Descriptive S-D logic concepts |
| Reference | [15,49,69–73] | [50–53] | [2,35,54,74–96] | [55–58,97–103] | [48,59–62,104–135] | [36,63–68,136–149] | Reference |

The aims will be achieved with two layers. Firstly, we need convincing guidance that has previously been successful at dividing numerous academic papers in various dimensions adjoining S-D logic philosophy; the theoretical continuum developed by Berghäll (2018) precisely fits our goal in this stage. In his paper, a three-dimensional classificatory schema is introduced to identify service marketing phenomena in a private forest owner's context from the perspective of S-D logic. This continuum casts light onto the value creation potential of the sector and reveals a clear gap in scholarly research: that more could be achieved by applying a service-dominant logic view. This successful attempt and findings inspire us to borrow the continuum in this paper, with the purpose of combining the S-D logic perspective with research found in the former section and trying to interpret them under the core sphere of S-D logic. Secondly, papers need to be graded based on a solid theory so their conceptual content will be transformed to quantities such as values, waiting to be further compared and visualized on each three-dimensional scale proposed by Berghäll. Here, we decide to borrow qualitative comparative analysis (QCA) to process papers. QCA is a comprehensive method that allows researchers to analyze data from a conceptual viewpoint [150], a set of techniques in which features of the case-oriented and variable-oriented strategies are combined [151]. QCA has been applied in several disciplines such as sociology, management studies, and political science [152]. It was initially intended to conduct small-N comparisons [153], and has been differentiated into three variations to suit various research aims, which are crispy-set QCA (csQCA) [153], fuzzy-set QCA (fsQCA) [154], and multi-value QCA (mvQCA), first proposed by Cronqvist (2003) [155]. The main difference between mvQCA and the other QCA-variants is that it allows multi-value conditions where each category is represented by a natural number (0, 1, 2, 3 . . . ), fsQCA allows every possible value between 0 and 1, while csQCA only allows 1 or 0 to indicate if a condition is present or absent [156]. Considering that our collected literatures are interdisciplinary in nature, and S-D logic is not an oversimplified existing or nonexistent phenomenon, we decided to apply mvQCA and assign papers with a value from 0 to 5 to indicate how much these papers fit in the S-D logic concept, value 0 representing comparatively no relation to S-D logic and 5 representing a high relation to S-D logic. MvQCA could process data further until researchers could understand what condition would be necessary and/or enough to explain sets; however, our aim by applying fsQCA is to divide papers into possible divisions so only the first step, "produce a data table", will be applied. Abstracts and keywords of 106 papers were examined again by one of the main authors, with the specific aim of positioning them on the proper scale. Table 2 depicts how we embed the descriptive S-D logic concept existing in six themes, shown in Table 1, into Berghäll's theoretical continuum, and the results of six themes with natural numbers from 0 to 5. The value is calculated by first coding each paper; then, taking the mean value of each theme, we round decimals up or down so the numbers fit the requirements of fsQCA.

Table 2 describes a continuum from AE as a product-dominant view to conceptualizations where AE is an interactive element of exchange. In this process, AE companies are value creators or a hindrance to this value creation. In line with Berghäll's continuum, this research applies "Logic of Value Creation" as the first axis to define the first dimension, which spans from the product-dominant logic view of the relationship to customer value creation and examines how close research is to approaching the A-to-A view of exchange. The second axis is "Logic/Model of Exchange", which refers to a unidimensional logic of exchange to dyadic exchange, finally approaching the systematic (contextual) view of the driven value chain of exchange. The third axis focuses on the content of exchange and would be a continuum from intangible to tangible, named as "The Definition of the Exchange Content".

**Table 2.** A comparative analysis of S-D logic concepts and literature.

| Themes | Logic of Value Creation (i.e., How Close to A-to-A) | Logic of Exchange | Exchange Content (i.e., How Close to Intangible) |
|---|---|---|---|
| History and Influence | Orientation towards A-to-A: 3 | How e-commerce influences agribusiness, unidimensional: 2 | Strive towards intangible, but focus more on tangible aspects: 3 |
| Performance | From AE performance evaluation slightly towards value creation understanding: 2 | Towards connections: 2 | Traditional product and website quality, strive towards intangible: 3 |
| Factors | Strive towards how to create co-values: 2 | Dyadic exchange: 2 | Satisfaction, clear towards intangible: 2 |
| Opportunities and Barriers | Towards value creation: 3 | Innovation, information exchange, social networks: 3 | Values, intangible contextual value creation/consumer understanding: 3 |
| Possible Solutions | From traditional marketing towards customer centricity (understanding customers' needs): 3 | Contextual and systematic: 3 | Clear intangible orientation, clearly point out value creation and information exchange: 3 |
| Consumer Willingness | Clear strive towards value creation/A2A: 3 | Contextual: 3 | Intangible: 4 |

## 6. Analysis of the Groups on S-D Logic-Axes

As can be seen from Table 2, the interactivity between S-D logic concepts and groups in the literature are quite different. It is necessary to point out that even though we coded papers with the same values, considering each theme may involve various dynamic concepts, their span could be dissimilar. Towards disclosure on how close the six themes approaching S-D logic concepts arrive in a visual way, the following content will present three figures separately. Figure 3 describes the results for the grading and grading classes in the Logic of Value Creation axis.

**Figure 3.** Logic of value creation.

As can be seen, at the end of the theme's spectrum, themes are clearly approaching A-to-A value co-creation. On a literature level, themes focus on Opportunities and Barriers, Possible Solutions and especially Consumer Willingness are gradually lay more attention on how to achieve consumers' satisfaction and how firms and consumers could integrate their resources to co-create value as a mutual service process. Wu (2018) [65] points out

that it is necessary for companies to understand customers' satisfaction, which directly leads the agricultural e-commerce to A2A direction in his research.

Under the theme of "Possible Solutions" and "Consumer Willingness", we find papers trying to give practical suggestions on how providers/companies could actively join into a value co-creation process by understanding consumer satisfaction and loyalty. These consist of, inter alia, through a high-level collaboration between customers and suppliers with online business management integration, supply chain management, and customer relationship management (CRM). For example, Alrousan and Jones (2016) [61] test factors affecting the levels of AE adoption among SMEs and then propose a model integrating customer service and CRM with companies' businesses online in Jordan. Wang and Tao (2021) [107] clearly mention the importance of value creation and propose "consumption-led business process model", "service-led organization model", "linear value creation model" to improve AE omnichannel supply chain and meet consumers expectation.

Figure 4 displays the axis of the logic of exchange, from information unidimensional to a systemic understanding of consumers' context.

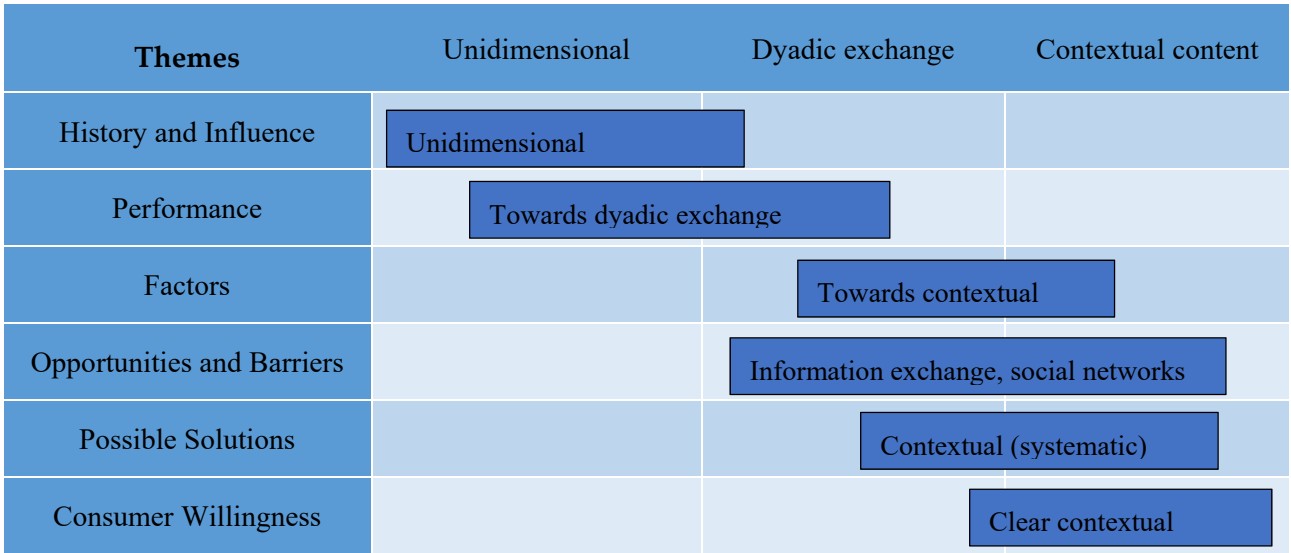

**Figure 4.** Logic of exchange.

As we can see from Figure 4, most of the research has an orientation towards contextual and systematic content. An exception is the papers in the History and Influence theme, which try to give a brief overview of AE's development without diving into multidimensional contents, thus being more unidirectional (see Yang et al., (2020) [69] as an example). The theme of Performance starts to consider cultural backgrounds and customs when evaluating AE performance; this theme therefore shows an attempt towards dyadic exchange (through agribusiness companies to consumers, vice versa), but still does not take the exchange context into contextual consideration (for example, advocating communities or third parties join in the exchange process). The theme of Opportunities and Barriers analyzes how social networks and information exchange could bring new opportunities to the AE and reach more consumers; examples are A. Dal Vecchio et al., (2018) [50], Borsellino et al., (2018) [51], etc. As for the theme of Consumer Willingness, all research takes consumers as the starting point and the center of the research, trying to understand consumers in their context, and this theme shows a dominant orientation in the exchange content.

Figure 5 presents the axis from tangible to intangible definitions of exchange.

| Themes | Tangible | Intangible |
|---|---|---|
| History and Influence | Food industry and B2B relationship, trust typology | |
| Performance | Traditional supply chain and consumer preference | |
| Factors | Satisfaction, towards intangible | |
| Opportunities and Barriers | | Values, consumer understanding |
| Possible Solutions | | Clear towards value co-creation |
| Consumer Willingness | | Highly towards intangible |

**Figure 5.** Exchange from tangible to intangible definitions of exchange.

Intangible exchange in our research refers to concepts related to consumer values, decision making, loyalty, consumer satisfaction, attitudes, perception, etc. We find that the theme of History and Influence, and Performance are in the transition stage, as research in each theme is nearly halved into both tangible and intangible content. For example, Canavari et al., (2010) [49] pay special attention to trust in a traditional B2B relationship and in an electronic context, while Baourakis et al., (2002) [15] introduce the impact that e-commerce has on the agri-food industry and marketing. Consumer Willingness leans strongly towards intangible exchange, because customer relationship management [64], customer satisfaction [65], loyalty, trust, etc., are largely discussed in this theme, but tangible resources, for example, web assurance seal services [66], and product and price [67], are still partly within the discussion. The Possible Solutions theme contains abundant intangible content such as political suggestions to encourage government participation [120], an optimization model of the agricultural products' distribution in an e-commerce channel [112], information sharing [117], etc. In the theme of Opportunities and Barriers, papers try to illustrate potential avenues and opportunities through intangible perspectives, for example, by using the SWOT method to describe AE strategies [100], legal regulation of digital development [102], etc. Papers as such bring this theme to a clear intangible orientation. The Factors theme shows more tangible manifestation as more papers concentrate on logistics processes [78], computer equipment [82], industry structure and product complexity [91], website usability [35] and so on.

Taken as a whole, several features emerge from three S-D logic axes presented in Figures 3–5. First, different themes of the AE literature perform unevenly on the three axes and show a disproportionality. The following two themes—History and Influence and Performance—mainly lay their focus on supply chain, the traditional agri-food sector, and similar product-dominant aspects, with comparatively less focus on consumer demands or multiple channel value exchange; therefore, these two themes are comparatively inferior on the axis of Value Exchange and axis of Value Creation. Themes such as Possible Solutions and Consumer Willingness are clearly approaching basic concepts of S-D logic on each axis. These two themes shift their attention to humanity elements, treating consumers as the major participants, seeking to offer solutions and to promote AE's efficiency from understanding consumers' demands, and at the same time, seeking companies' potential joint participation. In these two themes, consumer values and loyalty [63], and social network and multi-channel participation [120] are found frequently. One paper in the theme of Possible Solution directly borrows S-D logic to understand how digital commerce enables value co-creation in the vegetable supply chain in Indonesia [114]. We also find

another research work in the same theme that considers value creation and connects all parties in the supply chain to realize the information exchange [107]. The theme of Opportunities and Barriers introduces technical limitations [56] and likely opportunities existing in the AE domain in various countries, treating consumers as an inevitable element in the economic exchange process, even clearly pointing out that "Users create value by sharing and creating experiences in the web" [97] which shows this theme clearly approaches systematic value exchange and intangible content on the S-D logic axes.

Second, even though we discover that research is becoming closer to S-D logic premises, and ideologies that are increasingly approaching S-D logic have been applied to analyzing AE marketing strategies, policies, or to improve customer satisfaction, unfortunately, there only exists one research work that borrows S-D logic in the AE field, regardless of how influential S-D logic is in service and marketing disciplines and how many possibilities it could have functioning in the AE domain.

## 7. Discussion and Conclusions

### 7.1. Theoretical Implications

As e-commerce is gaining more and more attention as a domain, and its combination with agriculture increases AE's strong competitive strength, more and more research is appearing. Our systematic literature review collected 776 hints in the first searching trail, including research in Web of Science and Scopus. However, only 106 journal articles are left for further analysis when narrowing down our research focus closely with consumer, agriculture and e-commerce. Previous researchers have analyzed the AE phenomenon from various angles, for example, factors affecting AE performance, possible opportunities and barriers existing in AE domains; a certain set of solutions are given to guide future AE development. There are also plenty of articles putting their attention on consumers' choices, working on how to improve consumers' acceptance and satisfaction, etc. This orientation puts the consumer in a central position and tries to understand consumers in their context, what S-D logic has been reinforced as since its proposal. When comparing our research findings with other prior research that borrows S-D logic and conceptualizes consumers' collaborator role in various scenarios such as business systems and marketing practice [157,158], the highlights of this pioneering research are as follows. In this research, S-D logic serves as a basis for understanding value co-creation in a virtual agricultural e-commerce context. Through a comprehensive scanning of literature review work, we are able to investigate the locus, contribution, and lacking circumstances within this spectacular research scope, and how to reformulate our focus accordingly.

In detail, six themes occupied different percentages in the total research collected, for instance, the Performance theme only has 4 papers, History and Influence has 7 papers, Opportunities and Barriers (11 papers). By comparison, Consumer Willingness and Possible Solution have 21 and 37 papers, respectively, indicating that the research focus in the AE area is not even, the consumer theme and the solution theme being more popular recently. When combing S-D logic into the analysis of themes, various themes have a more unbalanced position on the three axes of S-D logic. For example, the theme of History and Influence along with Performance are inferior in the value creation axis and the logic of exchange axis, while Possible Solutions and Consumer Willingness are in the leading role on almost every axis of S-D logic. This indicates various AE themes still have gaps to fill to achieve a more systematic value exchange, especially those themes focusing on the AE phenomenon and tangible infrastructure.

Moreover, since S-D logic shows a powerful guidance in the AE field and applications are ready to launch, we would suggest that future research could apply S-D logic directly in this specific domain. Possible orientations might be: improving AE adoption from understanding the consumers context and meeting consumers' satisfaction; integrating companies' capabilities with social networks to strengthen companies' competitiveness; consumers as the central role in integrating intangible resources in the AE transaction process; connections and possibilities between AE and S-D logic, and so on.

*7.2. Managerial Implications*

From a managerial perspective, the research findings demonstrate the importance of understanding how the consumer is probing into participating in the interactive value co-creation process, and the promising opportunities waiting for marketers to initiate certain circumstances to co-create values together with their consumers. E-commerce, social media commerce, smartphone accessibility and digital technologies enabled consumers to actively engage with brands and create various types of user-generated content (UGC) [159]. It is insightful for AE service providers to meet consumers' willingness to better anticipate this collaborative process, and to encourage consumers to generate a favorable reputation of their brands. In here, we would like to introduce an excellent formula of embedding S-D logic into marketing that "The goal for marketing is to engage the supplier with significant customer practices and to contribute to value creation in those practices in a mutually beneficial way" [160], to shed a light on what the participants who are involved in the value creation process can do, to achieve maximum benefit. While this formula may originate from marketing, since the AE domain is interweaved considerably with the marketing section, this enlightenment may be applicable in a broader sense.

Due to its exploratory nature, this research has a number of limitations, which advocates the need to undertake further theoretical and empirical research in this emerging area. This study is situated in the agricultural e-commerce context and, thus, limits the researcher scope to explore S-D logic's application in other industry settings and might come up with partial conclusions. Meanwhile, in our database, the number of cases in each theme is not even and this may lead to bias when coding the data. We would suggest applying qualitative analysis, for example, cluster analysis, to guide the classifying process to avoid this uneven result. Finally, even though AE is becoming increasingly prevalent, this online trade prevalence does not imply the substitution of offline activity. Thus, there is a need for comparative research, which focuses on value co-creation in both offline and online settings.

**Funding:** This paper is funded by the NIEMI FOUNDATION in Finland (No. 20210026).

**Acknowledgments:** This paper is supported by the NIEMI FOUNDATION. Open access funding provided by University of Helsinki.

**Conflicts of Interest:** The authors declare no conflict of interest.

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
