# Peer review of "E-Commerce in Agri-Food Sector: A Systematic Literature Review Based on Service-Dominant Logic"

_jtaer, doi:10.3390/jtaer16070182_

Round 1

Reviewer 1 Report

The review prepared by the authors is a much needed and appreciated study into e-commerce in agri-food sector. The approach to the review based on service dominant logic brings an additional insight into the agricultural e-commerce. 

The paper is well organized and clearly specifies the path taken by the authors in conducting the study.

I do not see much need for corrections other than careful proofreading. I would also suggest that in the table 1 references should be presented as numbers as in the text to make it easier for the readers to look for the reference details at the end of the paper.

Author Response

Thank you for your comments.

This paper has been thoroughly read and edited by native English speakers for grammar and language correction this time.

Now the reference style in Table 1 is presented as numbers as in the text, also in line with the style of Journal of Theoretical and Applied Electronic Commerce Research.

Reviewer 2 Report

Dear authors,

Thank you for the possibility to read and evaluate your paper.
I am sending you my feedback in terms of Originality, Relationship to Literature, Methodology, Results, and Implications for research, practice and/or society.
My overall evaluation of the paper is a minor revision.

1. Originality
The theme of the article is actual, but the research questions are not surprising, and they confirm already known facts about e-commerce in the agri-food sector. The aim of the research is mentioned in the article as well as the research questions.

2. Relationship to Literature
The relationship to literature is quite well developed because many actual research papers from last years are covered. Also, the research gap is formulated in line with the previous researches.

3. Methods
The methodological part is described in a very logical way and brings detailed information on the sample selection algorithm

4. Results
The way of presentation of results is standard.

5. Implications for research, practice and/or society
The discussion should be more complex and provide readers with a comparison of the previous studies and your results. The theoretical implications and managerial implications are completely missing and the limitations of this study are vague. They should be definitely added to the current version of the manuscript.

Author Response

Thank you so much for your valuable comments.

1) A comparison of the prior research and the findings of this research is now added in the manuscript, the highlights and details are given.

2)Theoretical and managerial implications are illustrated now based on the research results.

3)Research limitations have been further explored and suggestions for future studies have been advocated as well.